# High Purity Struvite Recovery from Hydrothermally-Treated Sludge Supernatant Using Magnetic Zirconia Adsorbent

**DOI:** 10.3390/ijerph192013156

**Published:** 2022-10-13

**Authors:** Zhe Wang, Shuai Guan, Yajuan Wang, Wen Li, Ke Shi, Jiake Li, Zhiqiang Xu

**Affiliations:** State Key Laboratory of Eco-Hydraulics in Northwest Arid Region of China, Xi’an University of Technology, No. 5 South Jinhua Road, Xi’an 710048, China

**Keywords:** phosphorus, adsorption-desorption, hydrothermal treatment, magnetic zirconia

## Abstract

Recovery of phosphorus from sludge will help to alleviate the phosphorus resource crisis. However, the release of phosphorus from sludge is accompanied by the leaching of large amounts of coexisting ions, i.e., Fe, Al, Ca, and organic matter, which decreases the purity of sludge-derived products. In this study, an adsorption-desorption process using magnetic zirconia (MZ) as the adsorbent is proposed to obtain a high purity recovery product. The process involves selective adsorption of phosphate from the hydrothermally treated sludge supernatant (HTSS) using MZ, followed by desorption and precipitation to obtain the final product: struvite. The results indicated that at a dosage of 15 g/L, more than 95% of phosphorus in the HTSS could be adsorbed by MZ. Coexisting ions (Ca^2+^, Mg^2+^, Fe^3+^, Al^3+^, SO_4_^2−^, NO_3_^−^, Cl^−^, etc.) and organic matter (substances similar to fulvic and humic acid) in the HTSS had a limited inhibitory effect on phosphate adsorption. Using a binary desorption agent (0.1 mol/L NaOH + 1 mol/L NaCl), 90% of the adsorbed phosphorus could be desorbed. Though adsorption-desorption treatment, struvite purity of the precipitated product increased from 41.3% to 91.2%. Additionally, MZ showed good reusability, maintaining a >75% capacity after five cycles. X-ray photoelectron spectroscopy (XPS) indicated that MZ adsorbed phosphate mainly by inner-sphere complexation. This study provided a feasible approach for the recovery of phosphorus from sludge with high purity.

## 1. Introduction

As an essential element for food production, the demand for phosphorus has been rising sustainedly in recent years [1]. However, phosphorus resources are non-renewable, and global phosphorus resources with commercial value are expected to be depleted in the next 50–100 years [2]. Sewage treatment plants are an important phosphorus sink. Approximately 2.7 million tons of phosphorus enters sewage plants globally each year, representing 15% of total phosphorus emissions [3]. More than 90% of phosphorus in sewage ends up in sludge, where the mass fraction of phosphorus can be as high as 5–15% [4]. Therefore, the recovery of phosphorus from sludge would effectively alleviate the phosphorus resource crisis.

Sewage sludge contains pathogenic bacteria, microplastics, and heavy metals, thus its direct use as a fertilizer has been banned in many countries [5,6]. Therefore, to recover phosphorus in sludge, the release or solubilization of phosphorus from the waste solid is usually though to be the first step, though which phosphorus is separated from the co-existing hazardous materials [7]. Among various kinds of methods promoting the release of phosphorus from sludge, hydrothermal treatment at a relatively low temperature has been chosen as a priority technique due to its satisfactory phosphorus releasing effect and mild reaction condition [7,8,9,10,11,12,13]. Kuroda et al. reported that nearly all poly-phosphorus in activated sludge could be released under hydrothermal treatment at 70 °C for 1 h [8]. Xue et al. found that the release of orthophosphate in concentration is 90 mg/L when the treatment temperature and time were 50 °C and 1 h, respectively [9]. Similarly, a 51.6 mg/L of orthophosphate in the sludge supernatant was observed by Zhang and Kuba after hydrothermal treatment at 70 °C [10]. In addition, the release performance is usually intensified by adding acid-base reagents, since phosphorus bonded with metal ions and organic matter tend to be dissolved in acidic or alkaline environment [14,15,16,17]. Cheng et al. reported that compared to that under neutral conditions, the phosphorus release from sludge increased by 30% and 34% when the pH was adjusted to 2 and 12, respectively [17].

Both hydrothermal and acid-base treatment would cause simultaneous leaching of Fe, Al, Ca, and organic matter from sludge, lowering the purity of recovered phosphorus. Li et al. reported that the purity of recovered struvite was less than 20% when coexisting Ca^2+^ was more than 500 mg∙L^−1^ in wastewater [18]. In addition, Fe^2+^ and Al^3+^ impurities might result in the formation of non-bioavailable phosphorus in the recovered product [19]. Therefore, purification of the sludge derived phosphorus solution is required before recovery [7]. Na_2_S is added to the hydrothermally treated sludge supernatant (HTSS) to precipitate coexisting metal cations (by the reaction with S^2−^) to obtain a liquid containing only phosphorus. However, the addition of S^2−^ to the acidified HTSS may result in the formation of toxic H_2_S [20]. Adjusting the pH of the acid leachate of sludge to neutral allows phosphorus in the solution to be precipitated as Al–P and separated from the heavy metals that are still dissolved. However, subsequent conversion of Al–P to more readily bioavailable Ca–P is still required, thereby complicating the process [7]. In addition, coexisting ions and organic matter may be removed using nanofiltration and cation exchange resins, albeit at a high cost [21,22].

Selective adsorption of phosphorus from the HTSS followed by desorption to obtain a phosphorus-only solution is a possible route of purification [19,23]. Biswas et al. prepared Zr(IV) modified orange waste gel to adsorb phosphorus from an acid leachate of sludge incineration ash, resulted in efficient elimination of Ca, Fe, and Al [24]. Yu et al. used a Zirconium-containing material to selectively adsorb phosphorus from sludge leachate, followed by desorption and precipitation to obtain hydroxyapatite containing heavy metals (As, Cd, Cr, and Pb) at concentrations below the fertilizer safety standard values [6]. Lundehøj et al. investigated the adsorption performance of Zn_2_-Al-LDH on phosphorus in the acid leachate of sludge and found that Zn_2_-Al-LDH selectively adsorbed phosphorus from the leachate, but could not effectively eliminate SO_4_^2−^ [25].

Purification by adsorption-desorption is currently performed primarily in adsorption columns, which might suffer clogging during long-term operation [26]. Magnetic adsorbents can effectively resolve this issue. Magnetic adsorbents are dispersed in the liquid during adsorption process, and are magnetically separated after the reaction, thereby the clogging of adsorption column could be avoided [27]. Previously, we prepared a novel adsorbent, magnetic zirconia (MZ). The saturated magnetization of MZ is 23.65 emu/g, which enabled the magnetic separation after use [27]. The maximum P adsorption capacity is 42.19 mg/g, being higher or at least comparable with the recently reported Zr-containing adsorbent [26]. MZ exhibited good selectivity in the extraction of phosphorus from the secondary effluent of a sewage plant with a recovery rate higher than 90%. MZ could be repeatedly used in several cycles and its performance was rather stable [26,28]. Therefore, MZ is a promising adsorbent for the purification of the HTSS.

In this study, we investigated the selective adsorption of phosphate in the HTSS by MZ under acidic conditions and evaluated the purification effect of the adsorption-desorption treatment by testing the purity of the sludge-derived struvite. The adsorption-desorption mechanisms were also investigated though spectrum analysis.

## 2. Materials and Methods

### 2.1. Sludge Source

The excess sludge was taken from the secondary sedimentation tank of Xi’an No. 4 Wastewater Treatment Plant. Solid content of the raw sludge is 9.54%, which was higher than the reported average level of excess sludge [8,9]. So, it was diluted with distilled water to achieve a 5% solid content. The properties of the sludge mixture are presented in Table 1.

### 2.2. Hydrothermal Treatment

The hydrothermal treatment of sludge was performed in a glass flask placed in a water bath to control the reaction temperature. Mechanical stirring at 200 rpm ensured the uniformity. Based on the previous studies, the hydrothermal treatment was performed at 75 °C for 1 h [29]. To find the optimized pH for phosphorus release during the hydrothermal treatment, the pH of the sludge mixture was adjusted to 3, 4, 5, 6, 7, 8, 9, 10, and 11 with 1 mol/L of HCl and NaOH. After treatment, the sludge mixture was centrifuged at 300 rpm for 10 min, after which the total phosphorus (TP) and orthophosphate (PO_4_^3−^) concentrations in the hydrothermally treated sludge supernatant (HTSS) were determined using a colorimetric method. The solid residue of hydrothermal treatment at pH 3 was cleaned, dried, ground, and passed through an 80 mesh sieve; its phosphorus fractions were determined using the SMT method [15].

### 2.3. Preparation of MZ

FeSO_4_∙7H_2_O, FeCl_3_∙6H_2_O, and ZrOCl_2_∙8H_2_O were dissolved in 500 mL of ultrapure water to achieve a molar ratio of Fe^2+^/Fe^3+^/Zr = 1.33/2.66/1. The pH of the mixture was neutralized to 7.6 using 6 mol/L NaOH solution under mechanical stirring at 300 rpm. After aging at 60 °C for 16 h, the solid product was washed, dried, and sieved [26]. The obtained MZ was characterized by X-ray fluorescence (PW2404, PHILIP, Amsterdam, The Netherlands) and X-ray diffraction (D8ADVANCE, BRUKER-AXS, Berlin, Germany) to investigate its chemical composition and crystal phase. The magnetism and surface charge of MZ were measured by vibrating sample magnetometer (Lakeshore, Columbus, OH, USA) and zeta potential analyzer (BECKMAN COULTER, Brea, CA, USA).

### 2.4. Adsorption Experiment

A certain mass of MZ was weighed and placed in a 50 mL centrifuge tube, and 40 mL of HTSS was added. The HTSS was obtained under the following reaction conditions: treatment time of 1 h, reaction temperature of 75 °C, and initial pH of 3. The mixture was reacted at 180 rpm at 25 °C for a certain time and then centrifuged to determine the phosphate concentration in the supernatant. The calculation equations for phosphate adsorption and adsorption rate are given in Equations (1) and (2):(1)$$Q_{e}=\frac{\left(C_{0}-C_{e}\right)\times 40}{1000\times m}$$
(2)$$\eta =\frac{\left(C_{e}-C_{0}\right)}{C_{e}}\times 100\%$$
where *Q_e_* represents the phosphate adsorption amount (mg/g), *η* denotes the phosphate adsorption rate (%), *C*_0_ and *C_e_* represent the phosphate concentration in the supernatant before and after adsorption, (mg/L), and *m* denotes the mass of the MZ added (g).

In the dosage experiments, MZ was added at 0.1–0.8 g to attain the dosage of 2.0–20 g/L, and the reaction time was 48 h. In the kinetic experiments, 15 g/L MZ was added, and the reaction time was 0.5–48 h. In the coexisting ion effect experiments, solutions with single and multiple coexisting ions were prepared according to the concentrations of phosphate and coexistence ions, as well as pH in the HTSS (Table 2). The dosage and reaction time of MZ was 15 g/L and 48 h, respectively.

### 2.5. Desorption Experiment

MZ reacted with the HTSS at a dosage of 15 g/L, and the mixture was magnetically separated after 48 h. The phosphate concentration in the supernatant was measured, and the adsorption amount *Q_e_* was calculated according to Equation (1). Subsequently, 40 mL of desorption agent was added and reacted for 48 h. After the reaction, the phosphate concentration in the solution was measured by solid–liquid separation, and the desorption amount was calculated. The desorption amount and desorption rate were calculated according to Equations (3) and (4):(3)$$Q_{d}=\frac{C_{d}\times 40}{1000\times m}$$
where *Q_d_* represents the amount of phosphate desorbed (mg/g), *C_d_* represents the concentration of phosphate in the desorption solution (mg/L), and *m* represents the mass of MZ added (g).
(4)$$\eta^{\prime }=\frac{Q_{d}}{Q_{e}}\times 100\%$$
where *η*′ represents the phosphate desorption rate (%).

Unitary and binary desorption agents were used. The unitary desorption agents were NaOH and NaCl solutions with concentrations of 0.1, 0.5, and 1 mol/L. The binary desorption agents were mixtures of NaOH and NaCl with the following compositions: 0.1 mol/L NaOH + 0.1 mol/L NaCl, 0.1 mol/L NaOH + 0.5 mol/L NaCl and 0.1 mol/L NaOH + 1 mol/L NaCl. The concentrations of coexisting ions in the desorption solutions were also determined using ion Chromatography (Essentia IC-16, Shimadzu, Japan).

### 2.6. Recovery of Struvite from Desorption Solution

Phosphorus was recovered under the following conditions: Mg:P:N molar ratio of 1.2:1:1.6, pH = 9, reaction temperature of *T* = 25 °C, stirring rate of 200 rpm, and stirring time of *t* = 10 min [22]. MgCl_2_ and NH_4_Cl were used as the source of Mg^2+^ and NH_4_^+^. The pH of the desorption solution was adjusted with HCl and NaOH solutions. After precipitation, the obtained product was separated, then its crystal phase and the surface morphology were characterized using X-ray diffraction patterns (D8ADVANCE, BRUKER-AXS, Berlin, Germany) and scanning electron microscopy (SEM; HATACHI TM3000, Kyoto, Japan), and the phosphorus recovery efficiency (PRE) was calculated according to Equation (5):(5)$$R=\frac{C_{b}-C_{a}}{C_{b}}\times 100\%$$
where *R* represents the PRE (%); *C_b_* represents the phosphate concentration in the solution before recovery (mg/L), and *C_a_* represents the phosphate concentration remaining in the supernatant after recovery (mg/L).

To measure the struvite purity, the precipitate was dissolved by HCl, and the NH_4_^+^-N content of the solution was determined using ion chromatography (Essentia IC-16, Shimadzu, Kyoto, Japan). The purity of struvite product was calculated according to Equation (6) [18]:(6)$$P=\frac{C_{N}\times V\times M_{MAP}}{14\times m_{precipiate}}\times 100\%$$
where *P* represents the purity of struvite in the precipitate (%), *C_N_* denotes the concentration of NH_4_^+^-N in the product digestion solution (mg/L), *V* represents the volume of the precipitate digestion solution (L), 14 is the atomic weight of nitrogen (g/mol), *M_MAP_* is the molecular mass of struvite (g/mol), and *m_precipitate_* is the mass of the precipitate (g).

### 2.7. Reusability of MZ

The reusability of MZ was investigated in five continuous cycles of adsorption, desorption and regeneration experiments. MZ (0.6 g) and HTSS (40 mL) were added to a centrifuge tube and reacted at 25 °C and 180 rpm. After 48 h, the adsorbent was magnetically separated, and the remaining phosphate concentration in the supernatant was measured. The adsorption amount was calculated according to Equation (1). Upon washing with deionized water, the desorption agent was added. After 48 h of desorption, the adsorbent was magnetically separated, and the phosphate concentration in the desorption solution was determined. The amount of phosphate desorption was then calculated according to Equation (3). Regeneration was performed by adding 40 mL of deionized water to the adsorbent and adjusting the pH of the mixture to 4.5 using 0.1 and 0.01 M HCl. The adsorbent was then magnetically separated, and the supernatant was discarded, completing the regeneration. After regeneration, a new cycle of adsorption, desorption, and regeneration was conducted.

### 2.8. X-ray Photoelectron Spectroscopy (XPS) and Electron Dispersive Spectroscopy (EDS)

The adsorption/desorption mechanism was characterized by XPS (Axis Ultra, Kratos Analytical, Manchester, UK). XPS spectra were corrected using C 1s spectra (standard peak level 284.8 eV), and data processing and peak search were performed using Casa XPS software. The surface elemental compositions of unreacted MZ and MZ subjected to five adsorption–desorption–regeneration cycles were characterized using EDS (TM3000, Hitachi, Tokyo, Japan).

## 3. Results and Discussion

### 3.1. Hydrothermal Treatment

In order to find the optimized pH value during hydrothermal treatment for phosphorus release, phosphorus release performance at different pH values were investigated and the results are presented in Figure 1a. With the increase in pH from 3 to 11, the orthophosphate concentration in the HTSS decreased from 293.45 to 170.31 mg/L, and the corresponding release rate also decreased from 72.31% to 42.05%. In contrast, as pH increased, the TP concentration in the HTSS first decreased and then increased. The minimum TP concentration was observed at pH 7.

Such a tendency was observed primarily because the orthophosphate in sludge are mainly bound to the mineral fraction of the sludge, such as Ca, Fe, Al, Mg, and other (alkaline earth) metal elements, forming phosphate precipitates. These precipitates can dissolve at low pH, releasing the phosphate bound to them. On the contrary, they do not dissolve under alkaline conditions [15,25]. Thus, the higher release rate of orthophosphate under acidic pH could be seen. Organic phosphorus (OP) is primarily bound to organic matter in sludge. The organic matter releases under alkaline conditions, causing the simultaneous release of OP, which in turn leads to an increase in TP release when pH is higher than 8 [16,30]. Figure 1b presents the phosphorus fraction in sludge before and after hydrothermal treatment (pH 3). Evidently, hydrothermal treatment under acidic conditions induced the release of almost entire apatite phosphorus (AP, mainly in the forms of Ca–P) and of the vast majority of non-apatite inorganic phosphorus (NAIP, mainly in the forms of Fe–P and Al–P) in sludge, whereas only a small portion of OP was released [15].

Phosphorus recovery requires phosphorus to be in the form of orthophosphate whenever possible, so the optimal pH for hydrothermal treatment was 3 [9]. For the same reason, in the following part of the study, we mainly focused on the adsorption-desorption performance of phosphate by MZ. The concentrations of TP and phosphate in the HTSS were 311.79 and 293.45 mg/L, respectively, and the concentrations of other coexisting ions and chemical oxygen demand (COD_cr_) are listed in Table 2. Although the concentration of phosphorus was high, there were also significant amounts of coexisting ions in the HTSS, which would have a negative impact on the subsequent phosphorus recovery. Therefore, phosphorus in the HTSS needs to be further purified by adsorption–desorption treatment.

### 3.2. Adsorption of Phosphate in HTSS by MZ

#### 3.2.1. Characterization of MZ

The properties of MZ were characterized and the results are shown in Appendix A. The result of X-ray fluorescence indicated that MZ was mainly composed of ferric oxide and zirconium oxide, with a weight percentage of 21.8% of the latter (Appendix A). The crystal phase of ferric oxide in MZ was confirmed as magnetite by X-ray diffraction pattern (Appendix A). Nevertheless, the zirconium oxide might be amorphous since no related peak in the XRD pattern could be observed. The saturated magnetization of MZ is 23.65 emu/g, which was obtained from its magnetization curve (Appendix A). Zeta potential analysis showed that the surface MZ is variably charged and the isoelectric point is 9 (Appendix A), which favor anion adsorption under acidic and neutral enivironment.

#### 3.2.2. Effect of Dosage

The effect of MZ dosage on phosphate adsorption in the HTSS is presented in Figure 2a. The phosphate removal rate in the solution continuously increased with the increase of MZ dosage. Simultaneously, phosphate concentration in the liquid phase decreased. At the dosage greater than 15 g/L, the phosphate removal rate exceeded 95%, and the remaining phosphate concentration in the solution was below 20 mg/L. Although the phosphate adsorption performance could be further improved by increasing the dosage, the dosage of 15 g/L was chosen for the subsequent experiments out of economic considerations.

Figure 2b presents the phosphate adsorption amount of MZ at different dosages. The adsorption amount of MZ gradually decreased from 28.25 to 17.09 mg/g with increasing dosage. These values are significantly lower than those obtained in ultrapure water and domestic sewage (42.19 and 34.36 mg/g) in our previous studies, mainly because of the potential interference of the coexisting ions and organic matter in the HTSS [26,28]. However, phosphate adsorption amount of MZ under the present conditions were absolutely higher than several reported zirconium containing adsorbents, such as mesoporous zirconia [31], zirconium modified graphene oxide [32] and zirconium ferrites [33]. The relatively high phosphate binding capacity of MZ would benefit the completely recovery of phosphate from HTSS at a relatively low dosage.

#### 3.2.3. Adsorption Kinetics

The kinetics of the phosphate adsorption in the HTSS by MZ are presented in Figure 3a. Phosphate adsorption rate exceeded 80% at 30 min, after which it slowly increased, and finally reached an equilibrium at 1200 min with an adsorption rate of 98%. To clarify the effects of coexisting ions and organic matter in the HTSS on the adsorption kinetics, a control experiment was also undertaken in a simulated solution prepared according to the phosphate concentration and pH of the HTSS, and the results are presented in Figure 3b. Adsorption proceeded significantly faster in the control group, the adsorption rate reached 92% within 30 min, and the equilibrium reached at 240 min.

The adsorption kinetics in both the HTSS and the control solution showed that phosphate in the solution could be almost entirely adsorbed, indicating that the adsorption sites of MZ were sufficient. The differences in the adsorption kinetics were primarily attributed to the competitive effect of the coexisting ions (e.g., sub-Ca^2+^, Mg^2+^, Fe^3+^, Al^3+^) and organic matter in the HTSS, which inhibited the reaction [34]. The adsorption kinetics were fitted using the pseudo-first-order and pseudo-second-order models in Equations (7) and (8), respectively [35]:(7)$$ln\left(C_{0}/C_{t}\right)=K_{1}\cdot t+b$$
where *C*_0_ is the initial concentration (mg/L), *C_t_* is the concentration at time *t* (mg/L), and *K*_1_ is the kinetic constant ($\min^{-1}$).
(8)$$\frac{t}{q_{t}}=\frac{1}{K_{2}q_{e}^{2}}+\frac{t}{q_{e}}$$
where *t* is the contact time (min), *q_e_* is the adsorption capacity at adsorption equilibrium (mg/g), *q_t_* is the adsorption capacity at time *t* (mg/g), and *K*_2_ is the kinetic constant (g/(mg∙min)).

As presented in Table 3, r^2^ values for the pseudo-second-order model were higher than those for the pseudo-first-order model, indicating that the adsorption mechanism of MZ for phosphate is chemisorption [36].

#### 3.2.4. Effects of Coexisting Ions and Organic Matter

To clarify the effects of coexisting ions and dissolved organic matter in the HTSS on the phosphate adsorption performance by MZ, experiments were conducted in simulated solutions which were prepared according to the concentrations of phosphate and coexisting ions in the HTSS. The results were compared with the control (with only phosphate), and the reduction percentage in phosphate adsorption amount was calculated (Figure 4). Evidently, the competitive effect was limited in the presence of single ions, with the reduction percentage in adsorption amount was less than 5%. The inhibiting effect of SO_4_^2−^ was relatively higher, since zirconium hydroxide has a comparable affinity for SO_4_^2−^ than PO_4_^3−^ under acidic conditions [37]. The decrease in phosphate adsorption in the presence of multiple ions was smaller than for single ions, probably because of the antagonistic effects between different coexisting ions [38].

Notably, phosphate adsorption in the real HTSS was lowered by 10.77% compared to the control, which was significantly higher than in the simulated solutions with multiple kinds of coexisting ions. This indicates that other components in the HTSS, such as organic matter, also inhibited phosphate adsorption. The three-dimensional fluorescence distribution of the HTSS is presented in Appendix A, which indicated the presence of substances in the HTSS that were similar to fulvic acid and humic acid. According to the preliminary results, substances similar to both fulvic acid and humic acid could inhibit phosphate adsorption by zirconia [39]. Thus, the decrease in phosphate adsorption by MZ was the result of the combined effects of coexisting ions and organic matter. Admittedly, the inhibition effect of coexisting ions and organic matter was limited overall.

### 3.3. Desorption and Struvite Precipitation

#### 3.3.1. Desorption

After adsorption, MZ with adsorbed phosphate was magnetically separated and subjected to desorption treatment. The effect of two desorption agents, NaOH and NaCl, and their combinations on the desorption were investigated [36,40,41], and the results are presented in Figure 5. The desorption effect of NaCl was poor, with desorption rates between 6–8% regardless of the NaCl concentration. The desorption effect of NaOH was considerably better, and the desorption rate increased from 78.33% to 98.61% with its concentration increased from 0.1 to 1 mol/L. However, NaOH was more expensive than NaCl and might cause severe corrosion to the reactor [41]. Thus, binary desorption agents consisting of 0.1, 0.5, and 1 mol/L NaCl and 0.1 mol/L NaOH were further explored for its performance. It was observed that desorption rate exceeded 90% when using 0.1 mol/L NaOH + 1 mol/L NaCl as the desorption agent. Thus, 0.1 mol/L NaOH + 1 mol/L NaCl was used in subsequent experiments.

The content of coexisting ions in the desorbed solution was determined under optimal desorption condition, and the results are presented in Table 4. Concentrations of Ca^2+^, Mg^2+^, Fe^3+^, Al^3+^, and K^+^ decreased from hundreds of mg/L in the initial HTSS (Table 2) to less than 10 mg/L in the desorbed solution (Table 4). The concentrations of coexisting anions were also significantly lowered in the desorbed solution. The elimination rates of coexisting ions by adsorption-desorption treatment were calculated based on the concentrations of coexisting ions in the HTSS and the desorption solution, which is shown in Table 4. After the adsorption-desorption treatment, most of the coexisting ions in the liquid phase were successfully eliminated with eliminating rates ranging from 75 to 99.95%. A similar result was reported by Biswas et al., where zirconium modified orange gel was used as the adsorbent [24]. Generally, a purified phosphorus solution was thus obtained, laying a sound foundation for the subsequent recovery of phosphorus.

#### 3.3.2. Struvite Precipitation

Among different kinds of phosphorus recovery products, struvite stands out for its higher P_2_O_5_ weight percentage and bioavailability [12,42]. Struvite can be obtained by the addition of magnesium and ammonia salts to the desorbed solution at a certain molar ratio and under a certain controlled pH and reaction time. The results of XRD and SEM of the sludge-derived product obtained is presented in Figure 6. XRD pattern of the precipitated product (Figure 6a) exhibited strong peaks at 15.81, 20.85, 21.45, 31.91, and 33.67°, indicating the formation of magnesium ammonium phosphate, i.e., struvite crystals [43,44]. As shown in Figure 6b, the precipitation product was rod-shaped crystals, which was also consistent with the microscopic morphology of struvite crystals reported in previous studies [45]. Therefore, the main component of the precipitated product was confirmed to be struvite.

To clarify the effects of the adsorption-desorption treatment on phosphorus recovery efficiency, PREs in the HTSS and the desorption solution were calculated according to Equation (5) and the results are presented in Table 5. Evidently, the adsorption-desorption treatment did not have a significant effect on the recovery efficiency. However, the adsorption-desorption treatment did improve the struvite purity significantly. It was shown that struvite purity of the precipitate in the desorption solution was 91.3%, much higher than that precipitate in the HTSS. Compared with the struvite purities of products obtained by direct precipitation, fluidized-bed homogeneous granulation and seed crystallization, the adsorption-desorption method showed clear advantage [46,47,48]. Thus, adsorption-desorption treatment was an efficient way to get a purer recovery product from waste streams and sludge.

### 3.4. Reusability

To verify whether the MZ retains its performance during repeated use, a five-cycle adsorption–desorption–regeneration experiment was conducted, and the results are shown in Figure 7. After five cycles, MZ maintained 75% of the initial adsorption capacity, and the corresponding adsorption amount decreased from 23.37 to 17.56 mg/g. This indicates that the MZ is reusable in the treatment of HTSS. Using the binary desorption agent, the desorption rate in each cycle reached more than 90%, which not only effectively ensured phosphorus recovery but also vacated enough active sites for phosphate adsorption in subsequent cycles.

### 3.5. Mechanistic Study

#### 3.5.1. XPS Characterization

To elucidate the mechanism of adsorption-desorption of phosphate in the HTSS on MZ, the XPS spectra obtained before and after adsorption are shown in Figure 8. An outgoing peak at the binding energy of 133.9 eV was observed after adsorption (Figure 8a), indicating the effective adsorption of phosphate by MZ. Figure 8b presents the Zr 3d fine spectrum before and after adsorption. The binding energies of Zr 3d_3/2_ and Zr 3d_1/2_ orbitals were 182.39 and 184.77 eV before adsorption and 182.52 and 184.90 eV after adsorption. The change in Zr3d binding energy indicated the transformation in the chemical environment of zirconium atoms. Zirconium, as a transition metal element, has a hard Lewis acid property. Therefore, the zirconium atoms could form inner sphere complexes with phosphate anions via Lewis-acid-base interactions. The phosphate anions, which behave as a Lewis base here, donate electron pairs to the central zirconium atoms on the MZ surface [26,32,49]. The interaction could be expressed as the following Equation (9):(9)$$a\equiv ZrOH_{\left(s\right)}+H_{c}P{O_{4}^{c-3}}_{\left(aq\right)}+bH_{\left(aq\right)}^{+}⇋\equiv Zr_{a}H_{b}PO_{4\left(s\right)}+cH_{2}O_{\left(l\right)}+\left(a-c\right)OH_{\left(aq\right)}^{-}$$
where *OH* represents the hydroxyl group on the surface of zirconia, and *a*, *b*, and *c* are stoichiometric numbers.

Though the interaction, hydroxyl groups on the surface of MZ were replaced by phosphate anions, making the Zr-O-H linkages turned into Zr-O-P linkages. Since P has a higher electronegativity than H, the binding energy of zirconium atoms would shift to a higher value [32].

Equation (9) indicated that adsorption releases hydroxyl groups, and thus an increase in the concentration of OH^−^ in the solution can reverse the reaction, i.e., induce desorption [34]. Therefore, a sound desorption effect was achieved using NaOH solution as the desorption agent.

In addition, there should be a physical adsorption mechanism for phosphate adsorption on MZ. According to the result of Zeta potential analysis in Section 3.2, MZ is positively charged in the acidic HTSS. The main driving force of physical adsorption is the electrostatic attraction between phosphate and the positively charged MZ surface [50], expressed in Equation (10):(10)$$ZrOH_{2}^{+}+H_{2}PO_{4}^{-}⇌ZrOH_{2}^{+}-H_{2}PO_{4}^{-}$$

Phosphate adsorbed by electrostatic interaction could be desorbed in high-concentration electrolyte solutions, such as NaCl and KCl solutions [51]. In Section 3.3.1, the desorption rate was lower than 8% when NaCl was the desorption agent. This indicated that there was weak electrostatic adsorption of phosphate by MZ, and the contribution of physical adsorption is rather limited.

#### 3.5.2. EDS Characterization

The elemental composition of the MZ surface before the adsorption reaction was characterized using EDS, and the results are presented in Figure 9a. The main elements on the surface before adsorption were Fe, O, and Zr, along with small amounts of C and S. The latter two appeared mainly because of the pollution with CO_2_ in the air and FeSO_4_ used in the preparation of the adsorbent. The elemental composition of the adsorbent surface after five reaction cycles is presented in Figure 9b and Table 6. After five cycles, a significant increase in the elemental content of C was observed, Ca and Mg were also detected in this scenario. Atouei et al. found that metal hydroxides could adsorb Ca and Mg ions by forming inner sphere complexes, although their affinities toward the metal hydroxide surface are much lower than that of phosphate [52]. Li et al. reported that dissolved organic compounds in the liquid could be sequestrated by metal hydroxides though either inner sphere complexation or hydrogen binding [53]. The phenomenon here indicates that MZ, despite its high selectivity for phosphate in the HTSS, still adsorbed organic matter and coexisting ions. Moreover, the adsorbed coexisting ions and organic matter could not be further removed from the surface by desorption agents [39]. These substances were present on the surface of MZ and occupy some of the active sites, which may be the reason for the gradual decrease in the performance of MZ after several cycles.

#### 3.5.3. Summary of Mechanisms

MZ could adsorb phosphate in HTSS mainly though Lewis-acid-base interactions, i.e., the formation of inner sphere complexes, along with a minor contribution of electrostatic attraction. The Lewis-acid-base interactions could be reversed under the alkaline environment, while the phosphate anions bound though electrostatic attraction could be replaced by electrolytes, such as Cl^−^ and NO_3_^−^ anions. Therefore, the adsorbed P on MZ surface could be easily stripped by a combination of NaOH and NaCl desorption regent. Despite that MZ has a selectivity toward phosphate anions, it could sequestrate a small amount of co-existing Ca and Mg ions, as well as organic matter, which could not be desorbed by our desorption agents. The adsorption of these ions and organic matter has led to gradual deterioration of the MZ surface and occupied the active adsorption sites of MZ, lowering its capacity in the long run.

### 3.6. Implication for Phosphorus Recovery from Sludge

Hydrothermal treatment is recognized as a promising technology for the management of excess sludge in both volume reduction and nutrient-energy recovery [54]. In the present study, more than 70% of the phosphate in the sludge was liberated by hydrothermal treatment at 75 °C in 1 h. Compared with hydrothermal treatment or supercritical technology performing at higher temperatures and pressure, the present study is conducted under mild condition. Therefore, its energy consumption might be easily fulfilled by recovery of waste heat of wastewater [55]. Recently, energy recovery in waste water treatment plant (WWTP) using combined heat and power (CHP) schemes has drawn more and more attention, which might be another potential energy source for the hydrothermal treatment in the future [56].

In the present study, purification effect of the adsorption-desorption process was shown to be satisfactory in improving the struvite purity of the product. The feasibility of nutrient recovery from water or sludge supernatant using magnetic adsorbent has been explored for a long period of time. In our previous study, phosphorus recovery from secondary effluent of WWTP using MZ in a sequential batch reactor was investigated and favorable performance was observed [28]. Drenkova-Tuhtan et al. conducted a similar pilot scale test using a patented technique: high gradient magnetic filter as the adsorbent separator [57]. As for the gradual deterioration of the adsorbent in its repeated use, further modification of the MZ surface should be conducted in the future to avoid the harmful effect of co-existing ions and organic matter on phosphate selective adsorption.

To meet the challenge of phosphorus crisis, struvite recycled from waste water and sludge has become a substitution for the phosphorus rock in fertilizer production [2]. The European Union proposed the new version of Fertilizer Product Regulation in 2016, declaring that 30% of the phosphorus rock used in fertilizer production would be replaced by struvite recycled from WWTP [58]. The standard of waste water derived phosphorus fertilizer was also promulgated in the same regulation. For the better popularization of our recycling technique, the quality of the struvite obtained from sludge should be assessed using this regulation in the future.

## 4. Conclusions

An adsorption-desorption process for high purity struvite recovery from hydrothermally-treated sludge supernatant was proposed in this study. Chosen as the adsorbent, 15 g/L of MZ could selectively adsorb phosphate from the HTSS with a 95% removal rate. More than 90% of the adsorbed phosphate could be stripped using a binary desorption solution, and the struvite purity of the corresponding product was 91.3%, which was far higher than that of the product obtained directly from the HTSS. The adsorption mechanism was identified as the formation of inner sphere complexes, and the gradual decrease in the phosphate adsorption capacity of MZ during repeated use might be due to the slight and irreversible adsorption of coexisting ions and organic matter in the HTSS.

## Figures and Tables

**Figure 1 ijerph-19-13156-f001:**
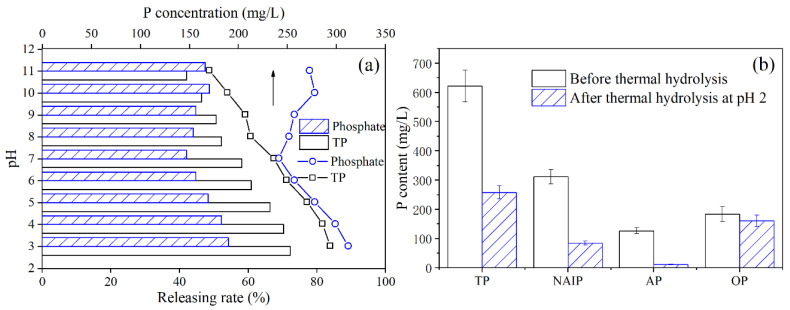
Effect of pH during hydrothermal treatment on P release (**a**) and change in P fraction in sludge before and after hydrothermal treatment at pH 3 (**b**). In (**a**), the lines and bars indicate the concentrations of phosphorus in the HTSS and the releasing rates, respectively.

**Figure 2 ijerph-19-13156-f002:**
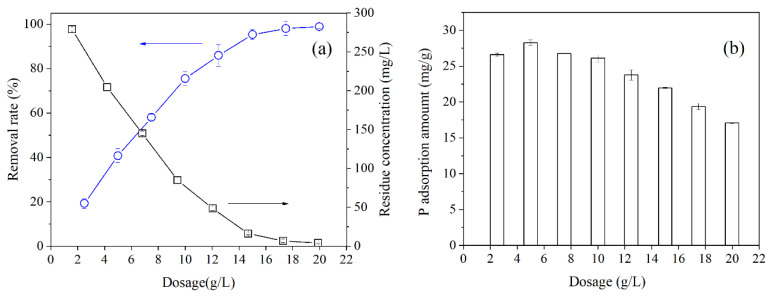
Effect of MZ dosage on phosphate removal rate, residue phosphorus concentration (**a**) and adsorption amount (**b**).

**Figure 3 ijerph-19-13156-f003:**
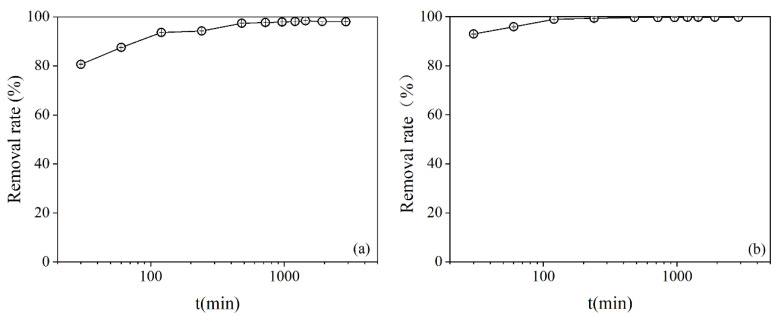
Adsorption kinetics in the HTSS (**a**) and control solution (**b**).

**Figure 4 ijerph-19-13156-f004:**
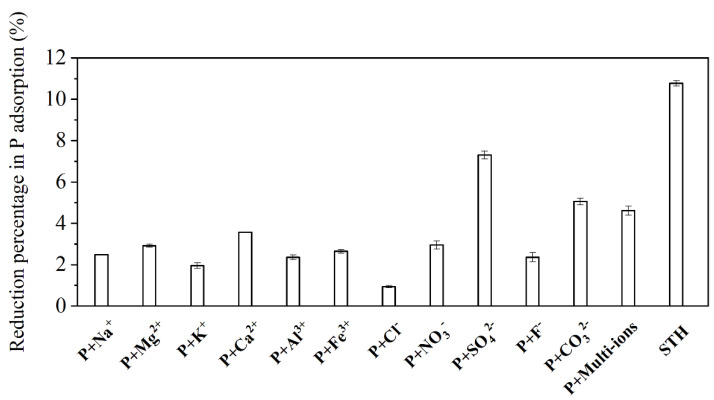
Effect of coexisting ions on the reduction of phosphate adsorption amount.

**Figure 5 ijerph-19-13156-f005:**
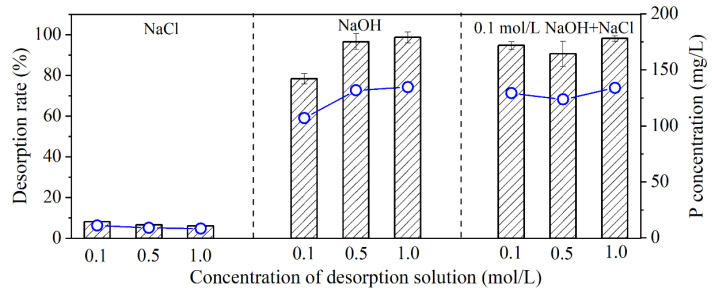
Desorption performance of different desorption agents. The bars indicated the desorption efficiency and the lines indicated the phosphate concentration.

**Figure 6 ijerph-19-13156-f006:**
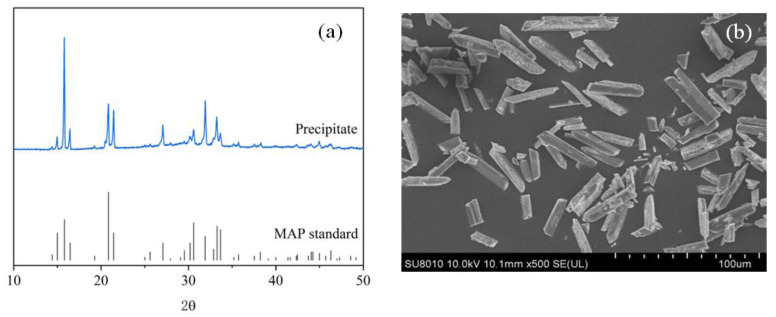
XRD pattern (**a**) and SEM images (**b**) of the precipitate.

**Figure 7 ijerph-19-13156-f007:**
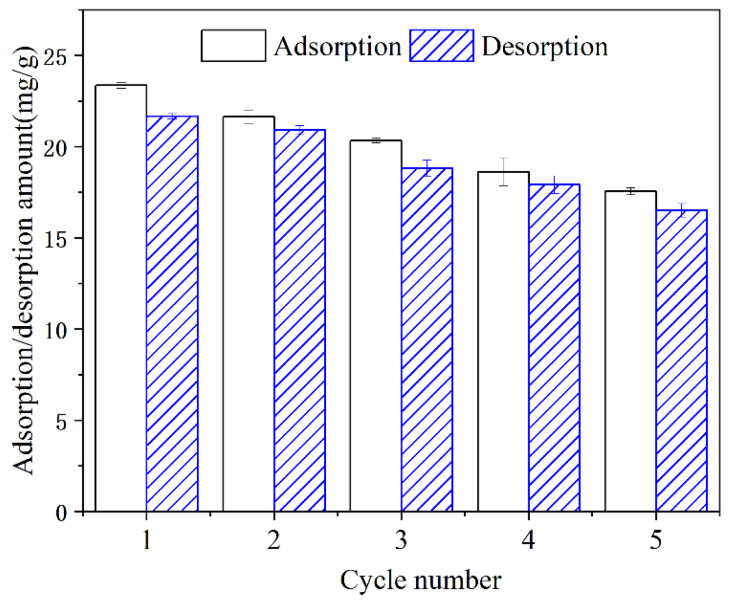
Adsorption and desorption performance in the cycling experiment.

**Figure 8 ijerph-19-13156-f008:**
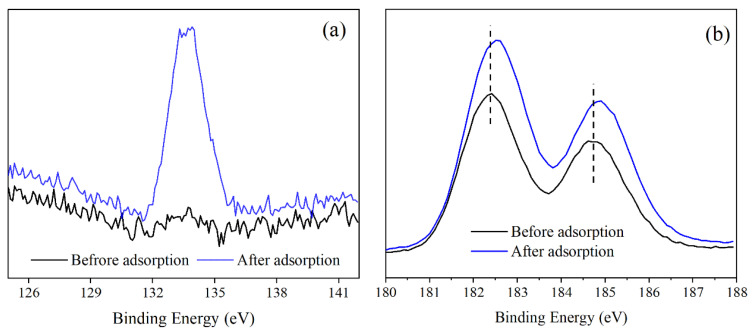
XPS P 2p (**a**) and Zr3d (**b**) spectra of MZ before and after adsorption.

**Figure 9 ijerph-19-13156-f009:**
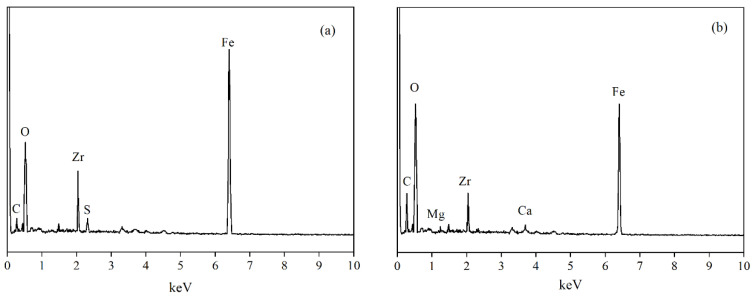
EDS spectra of MZ surface before adsorption (**a**) and after five cycles (**b**).

**Table 1 ijerph-19-13156-t001:** Basic parameters of sludge mixture.

Items	Units	Value
pH	-	6.9
TCOD	mg/L	46,505
SCOD	mg/L	41,200
TP	mg/L	573.20
PO_4_^3−^-P	mg/L	403.82

**Table 2 ijerph-19-13156-t002:** Concentrations of coexisting ions and COD_cr_ in HTSS obtained at pH 3.

Ion Species	Ca^2+^	Mg^2+^	Fe^3+^	Al^3+^	K^+^	Na^+^	F^−^	CO_3_^2−^	NO_3_^−^	SO_4_^2−^	Cl^−^	COD_cr_
Concentration (mg/L)	583.3	462.1	238.6	324.3	388.6	372.9	12.6	178.4	147.2	241.3	365.2	7625

**Table 3 ijerph-19-13156-t003:** Fitting results of adsorption kinetics.

Reaction Systems	Pseduo-First-Order Model			Pseduo-Second-Order Model		
	k_1_	q_e_	r^2^	k_2_	q_e_	r^2^
HTSS	0.07	1.05	0.60	0.45	22.68	1.00
Control	0.14	0.37	0.70	1.52	25.64	1.00

**Table 4 ijerph-19-13156-t004:** Coexisting ions and organic matter in the desorption solution and the corresponding elimination rate.

Ion Species	Ca^2+^	Mg^2+^	Fe^3+^	Al^3+^	K^+^	F^−^	CO_3_^2−^	NO_3_^−^	SO_4_^2−^
Concentration (mg/L)	2.44	0.21	10.31	0.56	4.83	2.10	14.51	36.55	42.66
Eliminating rate (%)	99.58	99.95	95.68	99.83	98.76	83.33	91.87	75.17	82.32

**Table 5 ijerph-19-13156-t005:** PRE and struvite purity of precipitate obtained from the HTSS and desorption solution.

Precipitate	PRE (%)	Struvite Purity (%)
HTSS	96.3	41.2
Desorption solution	95.8	91.3

**Table 6 ijerph-19-13156-t006:** Weight percentage of elements on MZ surface before adsorption and after 5 cycles (%).

Element	Fe	Zr	O	C	S	P	Ca	Mg	Al
Raw MZ	51.6	15.7	27	4.2	1.2	0.4	0.1	-	0.1
MZ after 5 cycles	38.2	9.8	38.3	10.5	-	1.6	1.1	0.4	0.4

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
