# Peer review of "High Purity Struvite Recovery from Hydrothermally-Treated Sludge Supernatant Using Magnetic Zirconia Adsorbent"

_ijerph, 2022, doi:10.3390/ijerph192013156_

Round 1

Reviewer 1 Report

This paper entitled: High purity struvite recovery from sludge thermal hydrolysate 2 using magnetic zirconia adsorbent, is an excellent paper and presents originality and usefulness to readers.

However, I regret that the authors did not characterize enough the adsorbent magnetic zirconia (MZ). Authors did not give any values on the Lewis acid base properties neither on the dispersive surface energy of the used materials.

On the other hand, the surface charge density and potential of the adsorbents can be very useful for the readers. If the authors have some information about this, it will be interesting to add it to the manuscript.

Author Response

Please see in the attachment.

Reviewer 2 Report

Comments on ijerph-1947043.

Title: High purity struvite recovery from sludge thermal hydrolysate using magnetic zirconia adsorbent

This study proposed an adsorption-desorption process using magnetic zirconia (MZ) to obtain high purity recovery product. The process involves selective adsorption of phosphate from the sludge thermal hydrolysate (STH) using MZ, followed by desorption and precipitation to obtain the final product: struvite. 15 g/L of MZ could selectively adsorb phosphate from the STH with a 95% removal rate. The adsorption mechanism was also investigated in this study.  The manuscript in current state is not suitable for publication unless they carefully address questions below. The main concern is about the term “thermal hydrolysis” used in this study.

Specific comments:

Abstract

1 Coexisting ions and organic matter in the STH had a limited inhibitory effect on phosphate adsorption. Please specify what kind of organic matter and ions here had inhibitory effect on phosphate adsorption.

2 Please summary your result in one sentence at the very end of the abstract. The significance of this study is also recommended.

Introduction

1 Line 36-37. Please double check your logical here. Why “Sewage sludge contains pathogenic bacteria, microplastics, and heavy metals, thus its direct use as a fertilizer has been banned in many countries” and then “Therefore, to recover phosphorus in sludge, the maximum release of phosphorus is imperative”?

2 I recommend to explain clear what is the “wet chemical approach”, please also pay attention to the terms throughout the manuscript when they are first mentioned.

3 Line 41-45, the introduction of THP here is so simple. I would ask why “thermal hydrolysis satisfactory phosphorus releasing effect and mild reaction condition”, and why “thermal hydrolysis is beneficial for subsequent sludge methanogenesis”, and why “thermal hydrolysis is usually intensified by adding acid-base reagents to achieve better releasing performance”? Please give the specific reasons between these facts such as what cases in the literature suggest that TH is beneficial for sludge methanogenesis and how, and under what condition?

4 Line 76, I still confused that why magnetic zirconia (MZ) was proposed here. In the literature, do you see any advantages of MZ over others? Please rewrite this part to make it clearer.

M&M

1 Line 86-89, please specify what kind of sludge used in this study. “waste sludge” is not the right term here. In addition, please explain why 5% solid content was used in this study.

2 section 2.2. I doubt that the TH described here is not the right TH. In the field of sludge treatment, TH is the process applied in wastewater treatment plants with anaerobic digestion. Thermal hydrolysis exposes sewage sludge or other types of wet organic waste to high temperature and pressure. Please modify this carefully, because this will mislead the researcher and practitioner.

3 Please explain why “The pH 3, 4, 5, 4 6, 7, 8, 9, 10, and 11 with 1 mol/L of HCl and NaOH” were tested here. I know pH might be an important parameter during TH, but general Ph in real sludge situation would not within that large range. I would recommend you careful explain the reason of this operation and give a reason if you wanted to investigate kind of mechanism.

Results and Discussion

1 please specify the unit for the concentration of irons listed in Table 2.

2 Line 213, if you want mean the organic matter is “COD” listed in Table 2, then please use COD.

3 Line 253, if you have replicate or triplicate, please provide the error bar in each figure.

4 Please carefully organize your results and discussion, most of time, the discussion is not enough. In addition, I recommend you to add an additional section to introduce the implication of this study in the very end of this study. May be some useful suggestions for both researchers and practitioner can be proposed based on your results. One example to modify your result is Line 373. “This indicates that the adsorption of phosphate by MZ mainly occurred through ligand exchange”. Please specify why the result from Figure 8(b) can support this conclusion, I mean, you have to tell us what the index from Figure 8b represent, and what information they can tell us. In addition, in the section of 3.5. Only described the results are far more enough. A summary for the mechanism is recommended.

Author Response

Please see in the attachment.

Round 2

Reviewer 2 Report

Authors have addressed all my questions. The manuscript can be accepted.